# Xylosyltransferase-Deficiency in Human Dermal Fibroblasts Induces Compensatory Myofibroblast Differentiation and Long-Term ECM Reduction

**DOI:** 10.3390/biomedicines12030572

**Published:** 2024-03-04

**Authors:** Anika Kleine, Matthias Kühle, Thanh-Diep Ly, Vanessa Schmidt, Isabel Faust-Hinse, Cornelius Knabbe, Bastian Fischer

**Affiliations:** Institut für Laboratoriums- und Transfusionsmedizin, Herz- und Diabeteszentrum Nordrhein-Westfalen, Universitätsklinik der Ruhr-Universität Bochum, Georgstraße 11, 32545 Bad Oeynhausen, Germany; mkuehle@hdz-nrw.de (M.K.); tly@hdz-nrw.de (T.-D.L.); vschmidt@hdz-nrw.de (V.S.); ifaust-hinse@hdz-nrw.de (I.F.-H.); cknabbe@hdz-nrw.de (C.K.); bfischer@hdz-nrw.de (B.F.)

**Keywords:** xylosyltransferase-I and -II, CRISPR/Cas9, extracellular matrix, GAG-linkeropathy, proteoglycan synthesis, glycosaminoglycan, myofibroblast differentiation

## Abstract

Desbuquois dysplasia type 2 (DBQD2) and spondylo-ocular syndrome (SOS) are autosomal recessive disorders affecting the extracellular matrix (ECM) and categorized as glycosaminoglycan (GAG) linkeropathies. Linkeropathies result from mutations within glycosyltransferases involved in the synthesis of the tetrasaccharide linker, a linker between the core protein of proteoglycan (PG) and GAG. DBQD2 and SOS are caused by the isolated mutations of the xylosyltransferase (XT) isoforms. In this work, we successfully generated *XYLT1*- as well as *XYLT2*-deficient GAG linkeropathy model systems in human dermal fibroblasts using a ribonucleoprotein-based CRISPR/Cas9-system. Furthermore, it was possible to generate a complete *XYLT*-knockdown. Short- and long-term XT activity deficiency led to the mutual reduction in all linker transferase-encoding genes, suggesting a potential multienzyme complex with mutual regulation. Fibroblasts compensated for ECM misregulation initially by overexpressing ECM through the TGFβ1 signaling pathway, akin to myofibroblast differentiation patterns. The long-term reduction in one XT isoform induced a stress response, reducing ECM components. The isolated *XYLT1*-knockout exhibited α-smooth muscle actin overexpression, possibly partially compensated by unaltered XT-II activity. *XYLT2*-knockout leads to the reduction in both XT isoforms and a strong stress response with indications of oxidative stress, induced senescence and apoptotic cells. In conclusion, introducing *XYLT*-deficiency revealed temporal and isoform-specific regulatory differences.

## 1. Introduction

The extracellular matrix (ECM) forms a dynamic network including a variety of structural proteins, such as collagens and proteoglycans (PGs). The latter play an important role in the homeostasis of the ECM, regulating hygroscopic abilities and, thus, influencing tissue elasticity [1,2]. They are composed of a core protein to which up to 100 different covalently bound glycosaminoglycans (GAGs) are attached. GAGs are unbranched polysaccharides consisting of repetitive, anionic disaccharides that can store a large amount of water due to their strong polyanionic charge [3]. The biosynthesis of the GAG chains starts with the formation of a tetrasaccharide linker that is O-glycosidically covalently bound to a serine of a consensus sequence in the core protein. The synthesis of the GlcA-β(1→3)Gal-β(1→3)Gal-β(1→4)xyl linker is catalyzed by five different glycosyltransferases. Firstly, xylosyltransferase (XT) transfers xylose during the rate-determining step. The enzyme has two isoforms, XT-I (MIM: 608124) and XT-II (MIM: 608125), which are encoded by the genes *XYLT1* and *XYLT2*. This is followed by the addition of two galactoses by galactosyltransferase-I (GalT-I; encoded by *B4GALT7*; MIM: 604327) and galactosyltransferase-II (GalT-II; encoded by *B3GALT6*; MIM: 615291). The linker region is complemented by the transfer of a glucuronyl acid via glucuronyltransferase (encoded by *B3GAT3*; MIM: 606374), which is followed by the polymerization of the respective GAG chains [4,5,6].

Linkeropathies are multisystemic genetic connective tissue diseases caused by the abnormal biosynthesis of PGs. This results in various clinical manifestations, such as short stature, skeletal deformities, brachycephaly, joint problems, facial dysmorphysm and cardiac defects [7]. One group are the GAG linkeropathies, which are characterized by mutations in the *XYLT1*, *XYLT2*, *B4GALT7*, *B3GALT6* and *B3GAT3* genes [8].

Desbuquois dysplasia type 2 is an autosomal recessive disorder characterized inter alia by severe pre- and postnatal growth retardation, short stature, joint hypermobility, joint dislocation and a flat face with prominent eyes. Genetic mutations in the *XYLT1* gene have been identified as the cause [9,10]. The mutations described in a total of 28 patients are heterogeneous and include splice, nonsense and missense mutations [9,10,11,12,13,14,15,16]. Another clinical picture associated with genetic changes in the *XYLT1* gene is the Baratela–Scott syndrome. The symptoms include short stature, patellar dislocation, shortened long bones and a flat face with a broad nasal bridge [17]. The causes are homozygous *XYLT1* mutations and the hypermethylation of the *XYLT1* gene [16].

Homozygous *XYLT2* mutations lead to spondylo-ocular syndrome, which is associated with short stature, retinal detachment, amblyopia, nystagmus, hearing loss, heart valve defects, bone fragility and mild learning difficulties [18,19,20,21]. To date, 24 patients from 13 different families have been described harboring this syndrome [18,19,20,21,22,23,24,25,26]. Predominantly homozygous mutations lead to the clinical picture of spondylo-ocular syndrome, but the most recently discovered pathogenic mutations were compound heterozygous [23].

The different clinical manifestations between Desbuquois dysplasia type 2 and spondylo-ocular syndrome indicate that XT-I and XT-II cannot compensate for each other and that there may be differences in the catalytic activity of the two isoforms [8]. Initial studies in model systems such as the mouse show reduced GAG concentrations and the resulting changes in PG expression due to mutations in the two XYLT genes [27,28]. The defective synthesis of the ECM may be the cause of short stature, the hyperflexibility of the joints and skin and abnormal wound healing. Due to the limited availability of patient material, the aim of this study was to establish CRISPR/Cas9-based knockout (KO) models of *XYLT1* and *XYLT2* in primary dermal fibroblasts. Primary cells offer the advantage of the most precise comparability with the original tissue. These in vitro models will be used to characterize the molecular mechanisms of GAG linkeropathies and contribute to the understanding of the crucial role of the XT isoforms in ECM homeostasis.

## 2. Materials and Methods

### 2.1. Cell Culture

Fibroblasts were purchased from Coriell (Coriell Institute for medical research, Camden, NJ, USA) and cultured in Dulbecco’s modified Eagles’s medium (Thermo Fisher Scientific, Waltham, MA, USA) mixed with 10% fetal calf serum (Gibco, Waltham, MA, USA), 2% L-glutamine and 1% antibiotic and antimycotic solution (100×; PAA, AT). Cells were cultivated at 5% CO_2_ and 37 °C.

### 2.2. CRISPR/Cas9-Based KO in Human Dermal Fibroblasts

The ribonucleoprotein-based CRISPR/Cas9 system was used to generate a KO of the XT genes. The complex consists of a Cas9 nuclease, an ATTO 550-labeled tracrRNA and a crRNA. Predesigned gRNAs from the manufacturer Integrated DNA Technologies (IDT, Coralville, IA, USA) were used (Table 1). An RNP complex was prepared for transfection. Subsequently, 11.5 μL crRNA (1 μM) and 11.5 μL of the Cas9 nuclease (1 μM) were added to OptiMEM (477 μL) and incubated (5 min, room temperature (RT)). The transfection reagent Lipofectamine 2000 (9.2 μL; Thermo Fisher Scientific, Waltham, MA, USA) was also diluted in OptiMEM (490.8 μL) and added to the RNP complex after incubation. The transfection mixture was placed into the wells of a 6-well plate and incubated again (20 min, in the dark). A total of 48,000 cells per cavity of a 6-well plate were resuspended in 2 mL antibiotic-free medium and dropped onto the transfection mixture. After 24 h, the medium was changed to medium containing antibiotic.

### 2.3. Fluorescence-Based Cell Sorting and Cell Separation

The cells were trypsinized at 24 h after transfection with the RNP-based CRISPR/Cas9 complex, and the cell pellet resuspended in Dulbecco’s phosphate-buffered saline (DPBS). Sorting was performed on the S3e Cell Sorter (BioRad Laboratories, Hercules, California, USA) by differentiation via emission at λ = 395 nm with excitation at λ = 561 nm. The cells obtained were transferred to collagen-coated 6-well plates and cultured to 90% confluence. In the next step, the cell pool was separated to generate single cell clones.

### 2.4. TA Cloning

TA cloning was conducted, according to the manufacturers protocol (Thermo Fisher Scientific, Waltham, MA, USA), to separate alleles and, therefore, to better characterize heterozygote mutations introduced by the CRISPR/Cas9 system. After the replication of the circular vector using E. coli TOP10, prokaryotic cells were lysed and plasmid DNA was purified and subsequently analyzed by sanger sequencing.

### 2.5. siRNA-Based Double-Knockdown (dKD)

Firstly, the working solutions (target siRNAs: 5 μM; control siRNA: 10 μM) of the siRNAs (Table 2) used were prepared from the stock solutions (50 μM) with nuclease-free water. The required number of cells were prepared in antibiotic-free medium according to the cultivation vessel. Regarding transfection, 10 μL Lipofectamine 2000 was diluted in OptiMEM (490 μL) and incubated (5 min, RT). The siRNA working solution (30 μL) was added to OptiMEM (470 μL) and then added to the Lipofectamine 2000 suspension and incubated (15 min, RT). The transfection mixture was added dropwise to the cell suspension. The medium was changed to a complete medium after 6 h.

### 2.6. RNA Extraction, Reverse Transcription and Quantitative Real-Time Polymerase Chain Reaction (qPCR)

The RNA was isolated using the NucleoSpin RNA kit (Macherey-Nagel, Düren, Germany), according to the manufacturer’s instructions. An amount of 1 µg of RNA was used for reverse transcription utilizing the SuperScript II Reverse Transcription kit (ThermoFisher, USA). The qPCR was performed using the LightCycler 480 II system (Roche, Mannheim, Germany). A reaction mix contained 2.5 µL cDNA (diluted 1:10 with H_2_O), 5 µL of SYBR Green Taq-Polymerase mix (Roche, Mannheim, Germany), 0.25 µL of each primer (25 pmol/µL; Table 3) and 2 µL H_2_O. Forty amplification cycles were used. The product was validated via melting curve analysis. Relative transcription levels were determined by the ΔΔC_T_ method.

### 2.7. Xyloslytransferase Activity Assay

The quantification of XT-I and XT-II activity was carried out according to the SPE-UPLC-MS/MS method of Kleine et al. [29].

### 2.8. Galactosyltransferase Activity Assay

The quantification of GalT-I was based on the SPE-UPLC-MS/MS method by Kuhn et al. [30].

### 2.9. Determination of the Sulfated GAG (sGAG) Concentration in Fibroblasts

The Blyscan sGAG assay kit (Biocolor Life Science Assays, Carrickfergus, UK) was used to determine the sGAG concentration in cell cultures. The assay was performed according to the manufacturer’s instructions.

### 2.10. Western Blot

The samples to be analyzed via Western blot were lysed with RIPA lysis buffer (Thermo Fisher Scientific, Waltham, MA, USA). A bicinchoninic acid assay was performed to determine the total protein concentration.

Regarding the detection of α-smooth muscle actin (α-SMA), 3.5 μg total protein solution was applied to 8–16% Tris-glycine gel. An amount of 5 μg total protein solution was separated using the 3–8% Tris-acetate gel for the detection of collagen type I (ColI). Separation was performed at 125 V for 2 h and subsequently transferred onto a polyvinylidene difluoride membrane using a semi-dry electro-blotting apparatus. The remaining binding sites of the membrane were saturated with a 5% BSA solution in TBS-T buffer for 1 h. Incubation with the primary antibody was carried out on the Coulter mixer overnight at 4 °C. The secondary antibody was incubated for 1 h at RT (Table 4). 

### 2.11. Bicinchoninic Acid Assay

A bicinchoninic acid assay was performed to normalize protein levels. The assay was carried out according to Smith et al. and performed as described previously [31,32].

### 2.12. Determination of the Pro-MMP1 Concentration in the Cell Culture Supernatants

The Human Pro-MMP1 Quantikine ELISA kit (R&D-Systems, Minneapolis, MN, USA) was used to quantify the Pro-MMP1 concentration in the supernatants of fibroblasts. The ELISA was performed according to the protocol provided by the manufacturer.

### 2.13. Determination of the TGFβ1 Concentration in Cell Culture Supernatants

The TGFβ1 concentration in cell culture supernatants was determined using the TGF beta 1 human ELISA kit (Abcam, Cambridge, UK). The ELISA was performed according to the protocol provided by the manufacturer.

### 2.14. Determination of Cell Viability and Caspase Activity in Fibroblasts

Cell viability can be determined using the ApoLive-Glo multiplex assay (Promega, Fitchburg, WI, USA) in a joint experiment with the caspase activity of fibroblasts. The assay was performed according to the manufacturer’s instructions.

### 2.15. Quantitative Senescence Assay

The fluorophore methylumbelliferyl-β-galactopyranoside (MUG) was converted to 7-hydroxyl-4-methylcoumarin by the senescence associated-β-galactosidase. The assay was carried out according to Schmidt et al. [33].

### 2.16. Determination of the Interleukin 6 (IL6) Concentration in Cell Culture Supernatants

The concentration of IL6 in cell culture supernatants was determined automatically on the Cobas e411 (Roche, Mannheim, Germany) using an electrochemiluminescence assay. Firstly, the samples were incubated with a monoclonal, biotinylated IL6 antibody. Another antibody directed against a different epitope, which is coupled to a ruthenium complex, was added and enabled magnetic fixation of the IL6 to the electrode with streptavidin-coated microparticles. After washing, the ruthenium complex was excited by applying a voltage and emitting light at a wavelength of 620 nm. The concentration of IL6 is proportional to the absorption.

### 2.17. Determination of the Intracellular Ca^2+^ Concentration of Fibroblasts

The intracellular determination of the Ca^2+^ concentration was made possible by the fluorophore Fluo-8-AM. The binding of Ca^2+^ ions increased the fluorescence of the molecule and, thus, enabled the quantification of the Ca^2+^ concentration within the fibroblasts.

Firstly, 8500 cells/well of a black 96-cavity plate (clear bottom) were seeded and cultured (24 h, 37 °C). The cells were first washed with DPBS and then overlaid with 100 μL Fluo-8 solution (15 μM in 0% fetal calf serum medium) (1 h, 37 °C) to load the cells with Fluo-8-AM. The cells were washed again (2×) and 100 μL 0% fetal calf serum medium was added. The fluorescence was measured every 30 s for 1 h at an excitation wavelength of λ = 490 nm and an emission wavelength of λ = 520 nm.

### 2.18. Inhibition of the TGFβ Receptor Kinase and Focal Adhesion-Associated Kinase (FAK) in Fibroblasts

In order to analyze the effects of TGFβ signal transduction, activin receptor-like kinase (ALK) 4, 5 and 7 were blocked by selective inhibition, thus impairing the signaling pathway. The inhibitor StemMACS SB431542 (TGFβRKI, 10 μM) was added to the medium for this purpose.

The FAK inhibitor PP2 (FAKI, 10 μM) was used to analyze the effects of signal transduction by FAK.

### 2.19. Immunofluorescence

In this study, the cells were cultivated in 8-well chamber slides, which were coated beforehand. The poly-L-lysine (100 μg/mL) coating was used to detect human ColI and the ColI (Rat tail; 35 μg/mL) coating was used for α-SMA. The fibroblasts were fixed either with methanol/acetone (1:1; for visualization of α-SMA) or with 4% paraformaldehyde (for the visualization of ColI) (15 min, 4 °C). Regarding permeabilization, the fixed cells were incubated with 0.1% Triton X-100 (10 min, RT), washed (3×) and incubated with 5% BSA in DPBS (1 h, RT). After another wash, the cells were incubated with the primary antibody (50 μL, 2 h, RT) and then washed and incubated with the secondary antibody (50 μL, 1 h, RT) (Table 5).

After nuclear staining with DAPI (300 nM, 5 min, RT), the cells were mounted with ROTIMount FluorCare (Roth, Mannheim, Germany). The analysis was performed by fluorescence microscopy.

### 2.20. Statistical Analysis

GraphPad Prism 9 (Boston, MA, USA) was used for the statistical analysis of this study. Data were analyzed using Mann–Whitney U test and are shown as the standard error of the mean (SEM).

## 3. Results

### 3.1. Generation of Separated CRISPR/Cas9-Mediated XYLT1 and XYLT2 KOs in Human Dermal Fibroblasts and Their Initial Characterization

Neonatal human dermal fibroblasts were transfected with an RNP-based CRISPR/Cas9 system (gRNA targets: *XYLT1* or *XYLT2*), sorted by flow cytometry based on the fluorescence signal of the RNP complex and then separated. A single clone colony with mutations was identified for each of the KO target genes *XYLT1* and *XYLT2* (Figure 1A). The *XYLT1* mutant clone showed overlaps in the genomic sequence from two bases upstream of the PAM sequence. No sequence overlaps were detected for the *XYLT2*-mutated clone. However, the comparison with the reference sequence showed a deletion of a thymine base three bases downstream of the PAM sequence. The TA cloning of the *XYLT1* genetically modified culture showed a heterozygous 5 bp deletion, which led to a frameshift with an early stop codon in the amino acid (AS) sequence. Only alleles with the 1 bp deletion, a homozygous mutation, could be detected for the *XYLT2* KO, and a resulting frameshift with a premature stop codon (Figure 1B). *XYLT1* mRNA expression was significantly decreased for both *XYLT1* and *XYLT2* genetically altered fibroblasts compared to the control (Figure 1C; 0.46-fold and 0.42-fold). The *XYLT1*-deficient culture showed no significant change in *XYLT2* mRNA expression compared to the control. The *XYLT2* mRNA expression of the *XYLT2*-mutated fibroblasts was decreased to 0.1-fold compared to the control fibroblasts (Figure 1C). The quantification of XT-I activity for both KO cultures showed a significant reduction to 0.45-fold of the control fibroblast activity. When XT-II activity was determined, no significant change was detected for the *XYLT1*-deficient fibroblasts compared to the control. By contrast, the *XYLT2*-deficient fibroblasts showed a significant reduction to 0.28 times the activity of the control (Figure 1D).

The *B4GALT7* mRNA expression analysis of the *XYLT1*-deficient fibroblasts showed a reduction to 0.72-fold of the control expression. The reduction in expression of the *XYLT2*-deficient fibroblasts was not significant. Both *XYLT1*- and *XYLT2*-deficient fibroblasts showed a significant reduction in *B3GALT6* mRNA expression to 0.89- and 0.83-fold of controls, respectively. A significant reduction in *B3GAT3* mRNA expression was again detected for both KO cultures analyzed compared to the controls (Figure 2A; 0.64- and 0.37-fold). When analyzing GalT-I activity, no significant differences were detected between the control and the *XYLT1*- or *XYLT2*-deficient fibroblasts (Figure 2B).

The altered gene expressions and enzyme activities of the transferases involved in the tetrasaccharide linker indicate an altered GAG synthesis. The sGAG concentration was significantly reduced for both *XYLT1*- and *XYLT2*-deficient fibroblasts to 0.62- and 0.73-fold of the control cell concentration, respectively (Figure 2C).

Changes in the GAG synthesis may have an impact on the concentration and functionality of PGs and the ECM homeostasis. The *XYLT1*-deficient fibroblasts showed a significant reduction in *DCN* (0.86-fold) and *ACAN* (0.07-fold) expression compared to controls. No significant changes could be detected for the expression of *COL1A1* and *MMP1*. By contrast, the *ACTA2* mRNA expression increased significantly to 1.60-fold of the control. The expression of *TGFB1* was slightly reduced to 0.73-fold of the control. The *XYLT2*-deficient fibroblasts also showed a significant reduction in *DCN* (0.50-fold) and *ACAN* (0.03-fold) mRNA expression compared to the control. No significant change was detected in *COL1A1* expression compared to the control. The *MMP1* mRNA expression showed a significant increase to 4.18-fold of the control expression. Significant reductions in the mRNA expression were detected in *XYLT2*-deficient fibroblasts for the genes *ACTA2* (0.47-fold) and *TGFB1* (0.64-fold) compared to controls (Figure 3A).

Western blot quantification showed a significant reduction (0.39-fold) in ColI expression in *XYLT1*-deficient cells compared to controls. The *XYLT2*-deficient fibroblasts showed a significant reduction to 0.07-fold of the control ColI expression (Figure 3B).

The Pro-MMP1 concentration showed a significant increase for both KO cultures compared to the control. The Pro-MMP1 concentration of the *XYLT1* KO fibroblasts increased 1.51-fold compared to the control concentration and the *XYLT2*-deficient cells showed a 4.32-fold increase compared to the control (Figure 3C).

Western blot quantification of α-SMA expression normalized to the respective GAPDH expression showed a significant increase to 3.43-fold of the control expression for the *XYLT1*-deficient fibroblasts. By contrast, the α-SMA expression of *XYLT2*-deficient fibroblasts was significantly reduced to 0.13-fold of the control expression (Figure 3D). 

The determination of the TGFβ1 concentration in the supernatants of *XYLT1*- and *XYLT2*-deficient cells showed no significant changes compared to the control (Figure 3E).

Finally, the effects of the altered ECM homeostasis on viability, senescence, apoptosis and oxidative stress were analyzed. The determination of the viability of the KO fibroblasts by means of aminopeptidase activity showed a slightly significant increase in both compared to the control of about 1.7-fold (Figure 4A). Quantification of SA β-galactosidase activity indicated a significant increase in the senescence of the *XYLT1*- and *XYLT2*-deficient fibroblasts relative to the control activity (Figure 4B; 2.68- and 1.5-fold). The analysis of caspase 3 and 7 activity, as markers of progressive apoptosis, showed a weakly significant increase to 1.3-fold over the control for the *XYLT2*-deficient fibroblasts (Figure 4C). 

The gene expression of *NOX4* and *IL1B* was significantly increased in the *XYLT1*-deficient fibroblasts (1.91- and 1.72-fold, respectively). The *IL6* mRNA expression showed a significant reduction to 0.73-fold of the control. No significant change could be detected for the *IL8* mRNA expression (Figure 4D). The *XYLT2*-deficient cells showed significantly increased levels of all four mRNA expressions compared to the control. *NOX4* showed a 2.58-fold induced expression, *IL1B* was increased to 67.77-fold of the control. The *IL6* showed a 4.03-fold overexpression and *IL8* was increased 285-fold compared to the control (Figure 4E).

The quantification of the IL6 protein concentration showed significant inductions for both KO cultures relative to the control concentration. The *XYLT1*-deficient fibroblasts were increased by 1.51-fold and the *XYLT2*-deficient fibroblasts showed 8.05-fold IL6 levels relative to controls (Figure 4F). 

The progression determination of the intracellular Ca^2+^ concentration using the fluorescence signal of Fluo-8 showed a greater increase in intracellular concentration for the *XYLT1*- and *XYLT2*-deficient fibroblasts compared to controls. The *XYLT2*-deficient fibroblasts reached a 2.14-fold concentration and the *XYLT1*-deficient fibroblasts reached a 1.46-fold concentration compared to the controls (Figure 4G).

### 3.2. Establishment of a Simultaneous XYLT1 and XYLT2 KO in Dermal Fibroblasts

A persistent double KO (dKO) of *XYLT1* and *XYLT2* was to be introduced into neonatal dermal fibroblasts to generate completely XT-deficient fibroblasts. Consequently, the fibroblasts were sequentially transfected with one RNP-based CRISPR/Cas9 complex each for *XYLT1* and *XYLT2*. The transfections were performed 24 h apart. Only 1.2 × 10^4^ cells could be detected in the *XYLT1*/*XYLT2* dKO fibroblasts 48 h after the second transfection with a CRISPR/Cas9 complex (Figure 5A; control: 2.3 × 10^5^ cells). The mRNA expression of the target genes *XYLT1* and *XYLT2* showed no changes in expression compared to the controls (Figure 5B). The DNA of the double-transfected fibroblasts showed a subsequence in addition to the main sequence in the area of the gRNA bindings for *XYLT1* and *XYLT2* (Figure 5C). However, this is hardly detectable but could indicate mutations in the cell pool. Overall, it was not possible to establish a stable and effective *XYLT* dKO in the fibroblasts.

### 3.3. Establishment of a Simultaneous XYLT1 and XYLT2 dKD in Dermal Fibroblasts

The generation of a persistent, complete XT-deficient KO model was not possible. A transient, siRNA-mediated dKD of *XYLT1* and *XYLT2* was performed in this subproject. Fibroblasts were transfected with *XYLT1* and *XYLT2* siRNA (50 nM each) to establish the dKD of *XYLT1* and *XYLT2*. The *XYLT1* mRNA expression of the double-deficient fibroblasts showed a significant reduction in the control mRNA expression to 0.18-fold. A significant reduction in *XYLT1*/*XYLT2* dKD fibroblasts to 0.14-fold compared to the control was detected for the relative mRNA expression of *XYLT2*. The method established here for the lipofectamine-mediated dKD of *XYLT1* and *XYLT2*, thus showing a knockdown efficiency of >80% for both target genes (Figure 6A). The *B4GALT7* mRNA expression of *XYLT1*/*XYLT2* double-deficient fibroblasts decreased significantly to 0.74-fold compared to control fibroblasts. No significant change was detected in the mRNA expression of *B3GALT6* between the controls and the dKD fibroblasts. The relative *B3GAT3* mRNA expression of fibroblasts was significantly reduced by the *XYLT1*/*XYLT2* dKD to 0.51-fold of the control (Figure 6A). The *XYLT*-deficient cells showed a significant reduction in the XT-I activity to 0.05-fold of the control activity. The XT-II activity of the *XYLT*-deficient fibroblasts was significantly reduced to 0.19 times the activity of the controls (Figure 6B). No difference in the GalT-I activity was detected between the controls and the *XYLT1*/*XYLT2* double-deficient fibroblasts (Figure 6C).

The ECM homeostasis was investigated after analyzing the expression of the enzymes involved in tetrasaccharide linker synthesis. The mRNA expression of the two PG genes *DCN* and *ACAN* was significantly increased in the *XYLT1*/*XYLT2* double-deficient fibroblasts compared to the control (Figure 7A; 1.38- and 4.09-fold). The expression of *COL1A1* was also significantly increased 1.48-fold in the dKD cells compared to the control. In line with this, the *MMP1* mRNA expression of the *XYLT1*/*XYLT2*-deficient fibroblasts decreased significantly to 0.66-fold of the control expression. The *ACTA2* and *TGFB1* genes were each significantly increased in expression in the dKD fibroblasts (Figure 7A; 1.70- and 2.20-fold, respectively).

The mRNA expression analysis revealed a variety of changes in the ECM homeostasis. Some of the targets analyzed were subsequently characterized at the protein level. The quantification of the fluorescence signal indicated a significantly increased ColI expression of *XYLT1*/*XYLT2*-deficient fibroblasts compared to controls (Figure 7B; 1.23-fold). Figure 7B shows exemplary images of the control and dKD fibroblasts at 100× magnification. The extracellular Pro-MMP1 concentration in the *XYLT1*/*XYLT2*-deficient fibroblasts was significantly reduced to 0.36-fold of the Pro-MMP1 concentration in the control cells (Figure 7C). The quantification of the fluorescence signal indicated a significantly increased α-SMA expression of *XYLT1*/*XYLT2*-deficient fibroblasts compared to controls (Figure 7D; 2.13-fold). Figure 7D shows exemplary images of the control and dKD fibroblasts at 100× magnification. The α-SMA could be detected in fiber-like structures in the morphologically magnified dKD cells. The extracellular TGFβ1 concentration of the *XYLT1*/*XYLT2*-deficient fibroblasts was significantly increased to 2.75 times the TGFβ1 concentration in the control cells (Figure 7E).

### 3.4. Analysis of TGFβ Signal Transduction following XYLT1/XYLT2 dKD in Dermal Fibroblasts

The *XYLT1*/*XYLT2* dKD showed an induction of ECM protein expression, which is characteristic for myofibroblast differentiation. The TGFβ signal transduction can trigger the differentiation of fibroblasts into myofibroblasts. TGFβ induces the de novo synthesis of α-SMA and, thus, the formation of the myofibroblast-typical phenotype and a feed-forward reaction of further cytokine production [34]. The signaling pathway was inhibited to analyze the causal relationship between TGFβ1 induction and myofibroblast differentiation in *XYLT1*/*XYLT2* dKD fibroblasts. The specific inhibition of the kinase domain of the TGFβ receptor was investigated using SB431542 (TGFβRKI). In addition, the signal transduction via the FAK, which is also relevant for myofibroblast differentiation, was inhibited by the specific inhibition of the Scr domain using PP2 (FAKI) (Figure 8A). 

The *XYLT1*/*XYLT2* double-deficient fibroblasts showed an increase (1.84-fold) in α-SMA expression compared to the control. Cultivation with the addition of TGFβRKI showed no significant difference in α-SMA expression between the control cells and the *XYLT1*/*XYLT2* dKD fibroblasts. The addition of FAKI during the cultivation of control fibroblasts and dKD cells also resulted in no significant difference between the two conditions in terms of the α-SMA expression. The comparison of the α-SMA expression of the non-inhibited *XYLT1*/*XYLT2* dKD fibroblasts (−I) with the α-SMA expression of the inhibition approaches of the dKD cells (TGFβRKI or FAKI) showed a significant reduction (0.51- and 0.61-fold, respectively) in each case (Figure 8B).

Images of fibroblasts cultured without an inhibitor (−I) showed α-SMA-positive cell bodies and the formation of fiber structures after *XYLT1*/*XYLT2* dKD. The corresponding control showed isolated weak α-SMA-positive cell bodies. The sample images of the controls treated with the respective inhibitors (+TGFβRKI, +FAKI) also showed hardly any α-SMA-positive cells. The *XYLT1*/*XYLT2* dKD fibroblasts treated with the respective inhibitors also showed only a few α-SMA-positive cell bodies (Figure 8C).

## 4. Discussion

### 4.1. Characterization of the Introduced Genetic Mutations 

An RNP-based CRISPR/Cas9 system was used to generate the desired KOs. The components required for the CRISPR/Cas9 system were not overexpressed by a plasmid or viral vector within the cell but transfected in a limited concentration as an RNP complex. This limitation reduces the possibility of off-target effects [35,36]. A further advantage of the RNP system is the degradation of the components within about 24–48 h, which can increase the viability of the transfected cells [37]. This is a particularly important factor in the transfection of sensitive primary cells.

A heterozygous clone was identified for the *XYLT1* KO system. The 5 bp deletion upstream of the PAM led to a reading frame shift and the resulting stop codon (p.Leu289Serfs*4). The resulting protein has only 292 instead of 959 AS and ends at the beginning of the transmembrane domain without the catalytic region of the protein. It is known from the literature that truncated XT-I proteins can lead to delocalization [9,11,15]. The mutation p.Arg481Trp leads to a ubiquitous distribution of the protein in the cytoplasma instead of localization in the Golgi, and the mutation p.Val724Serfs*10 allowed the truncated XT-I protein to be found in the endoplasmic reticulum. The mutant detected in this work is shorter than the delocalized variants and, therefore, possibly also localized in the cytoplasma. Incomplete *XYLT1* mRNA is, thus, not necessarily completely degraded by the nonsense-mediated mRNA decay mechanism but can also lead to mislocalized proteins. An accumulation of nonfunctional proteins, for example in the endoplasmic reticulum, is a possible trigger of cellular stress responses [38,39].

The *XYLT2* KO culture analyzed showed a homozygous 1 bp deletion, which led to a premature stop codon and, thus, significantly shortened XT-II protein (98 instead of 865 AS). The literature also describes almost exclusively homozygous *XYLT2* mutations, which lead to variable phenotypic characteristics [19]. The mutation described here is located at the beginning of the transmembrane domain and, thus, upstream of the catalytic domain [19]. The resulting protein can presumably neither be correctly localized nor catalytically active.

The two most probable off-target regions for both KO systems generated were analyzed by Sanger sequencing and no changes in the genomic sequences within these regions were detected. The off-target sequences with the most sequence homologies had at least three mismatches. Base mismatches in the gRNA region lead to a strongly reduced efficiency of Cas9 [40]. The effects observed can, therefore, be attributed to the mutations in the target genes.

### 4.2. Analysis of Tetrasaccharide Linker Transferase Expressions and Their Activities

The *XYLT1* KO cells showed significantly reduced *XYLT1* mRNA expression (to 46%). The *XYLT2* KO also leads to reduced *XYLT2* expression (to 10%) in the cells. The heterogeneous character of the *XYLT1* KO may explain the higher residual expression of *XYLT1* mRNA. The remaining wild-type allele can still be fully expressed. The literature also shows a higher residual expression in heterogeneous clones compared to two mutant alleles [41]. The *XYLT1* KO has no effect on the mRNA expression of the *XYLT2* gene. Fischer et al. previously detected an induction (to 130%) of *XYLT2* mRNA expression in a *XYLT1*-deficient fibroblast model [42]. The *XYLT2* KO, on the other hand, leads to a reduction in *XYLT1* mRNA expression (to 42%). Changes in *XYLT1* mRNA expression in *XYLT2* deficiency have not yet been investigated in other models. The analysis of XT isoform activities confirmed the observations described at the mRNA level. The literature also shows reduced XT activities with mutations in the two genes [27,42,43]. The method of isoform-specific activity determination has only been in use since 2023, which is why no literature comparisons can be used [29]. This work, therefore, shows the different regulations of the two XT isoforms for the first time. The XT-I deficiency results in suppression of XT-I expression and activity but not in any altered expression of XT-II. The XT-II KO, on the other hand, regulates both XT-I and XT-II expression and activity. The two isoforms appear to be differentially regulated.

### 4.3. Analysis of sGAG Concentration and DCN and ACAN mRNA Expressions

The altered gene expressions and enzyme activities of the transferases involved in tetrasaccharide linker synthesis suggest altered sGAG synthesis. Altered GAG synthesis may also have an effect on PG synthesis [6]. There is a reduction (to 62% and 73%, respectively) in the sGAG concentration in both *XYLT1*- and *XYLT2*-deficient fibroblasts compared to control cells. This can be explained by the reduced XT activity as well as the reduced mRNA expression of the downstream transferases. Without a functional tetrasaccharide linker, no disaccharides can be attached to synthesize GAGs [6,44]. *XYLT1*-deficient cells exhibit reduced sulfate incorporation in PGs, which is due to reduced GAG production [27,45]. A general reduction in GAG concentration was also observed in *mXylt2*-deficient mice [28,46].

Reduced GAG concentrations can also be detected for other GAG linkeropathy models. A reduction (to 50%) was demonstrated in *B4GALT7*-deficient zebrafish using the Blyscan assay [47]. Malfait et al. were able to detect a reduction in the sGAG concentration for *B3GALT6*-deficient patient fibroblasts, and a *B3GALT6* KO system in zebrafish also showed a reduced GAG concentration [48,49,50]. The GAG analyses of various *B3GAT3* mutations from patient material and in the zebrafish model also show reduced HS and CS levels and, thus, reduced GAG concentrations [51,52,53,54].

*ACAN* expression is very strongly repressed in both KO cultures (to 3–21%). Fischer et al. were unable to detect *ACAN* expression in fibroblasts after a *XYLT1* KO, nor were they able to induce it by adding TGFβ1 [42]. There is evidence in the literature of a correlation between reduced *ACAN* mRNA expression and short stature in patients [55,56]. Short stature is a prominent feature in all GAG linkeropathies. Aggrecan is an important component of cartilage particularly, together with collagen type I. The aggrecan aggregates form a large network that enables the shock resistance of the tissue by binding water molecules [57]. Osteoarthritis is a joint disease in which there is a loss of PGs and GAGs in the cartilage. Reduced *XYLT1* mRNA expression and associated *ACAN* mRNA expression were detected in a rat model [58]. The exact relationships between the regulation of *ACAN* expressions have not yet been discovered. However, it has been shown that the aggregates are degraded by proteases, such as MMPs, and that this degradation can be promoted by cytokines [57]. In addition, cytokines, such as IL1β, repress the expression of *ACAN* [57,58]. In line with the reduction in *DCN* mRNA expression detected in this study and an even more extensive reduction in *ACAN* expression, Han et al. have already shown a negative regulation of *ACAN* expression by a *DCN* KO [59]. This could be one reason for the strikingly strong regulation of *ACAN* expression in the cultures analyzed in this study.

### 4.4. Analysis of the Expression of ECM-Associated Proteins

Without a fully synthesized tetrasaccharide linker, PG biogenesis cannot be terminated, which, for example, disrupts functional ColI fibril formation [60]. Reduced ColI protein concentrations could be determined for the two *XYLT* KO fibroblast cultures by Western blot analysis. This observation was also detected by Fischer et al. in a *XYLT1*-deficient fibroblast model [42]. Nonfunctional ColI fibrils, for example, are recognized by the cell and degraded by MMP1 [61]. An induction of the pro-MMP1 concentration in the supernatants of *XYLT1*- or *XYLT2*-deficient fibroblasts can be observed in this work. This altered collagen synthesis is linked to various skeletal dysplasia. Defective collagen synthesis is described in the literature as the cause of a large number of subtypes of Ehlers–Danlos syndrome. Symptoms such as joint hyperflexibility, abnormal wound healing and hyperextensible skin occur [60,62,63,64].

It can be assumed that the mechanical properties of the ECM are disturbed by the defective GAG synthesis and the associated alteration of ECM synthesis and that signal transductions comparable to a wound injury could possibly be triggered. It is possible that TGFβ1 can no longer be stored in the ECM structure [65]. The altered ECM homeostasis is also likely to affect the elasticity and tensile strength of fibroblasts and have an impact on their signal transduction. The increased firmness of the surrounding ECM may trigger myofibroblast differentiation [66,67,68]. The mutant fibroblasts analyzed exhibited large and sprouting cell bodies. The α-SMA is a marker protein for the differentiation of fibroblasts into myofibroblasts [69,70]. The *XYLT1*-deficient fibroblasts expressed significantly increased levels of α-SMA compared to their controls (343%). By contrast, the *XYLT2* KO cells showed a reduction in α-SMA to below 15%. The observations regarding the *XYLT1*-deficient culture were also detected by Fischer et al. [42]. However, Fischer et al. also described patient fibroblasts with extensive genomic deletion, including *XYLT1*, which exhibited an α-SMA reduction [43]. The *XYLT1*-deficient fibroblasts appear to be able to produce α-SMA. However, no feed-forward loop is initiated, which triggers TGFβ expression and generally increased ECM production. The ECM production of the *XYLT2*-deficient fibroblasts is completely shut down. The nevertheless altered phenotype of the *XYLT2*-deficient fibroblasts can possibly be explained by cell senescence [71]. The two cell states are closely linked. Myofibroblasts also pass over into senescence after the wound closure is complete. 

### 4.5. Analysis of Viability, Senescence, Apoptosis and Oxidative Stress

The mutations introduced in the XT genes lead to homeostasis changes in the ECM. The viability of both KO models is significantly induced. The modified mRNA transcripts and possibly misfolded and mislocated proteins can trigger stress responses, such as the nonsense-mediated mRNA decay and the unfolded protein response in the fibroblasts analyzed [38]. The increased aminopeptidase activity could be due to an induction of the unfolded protein response and the endoplasmic reticulum-associated protein degradation. These rescue mechanisms of the cell recognize misfolded and accumulating proteins and degrade them [72,73]. For this purpose, the synthesis of chaperones and other auxiliary proteins is stimulated. An altered metabolic profile can also be the reason for increased aminopeptidase activity. The analysis of SA β-galactosidase activity shows an induction for both KO models. The induction of the cytokines of the senescence-associated secretory phenotype accompanies the entry into senescence and changes the aminopeptidase activity of the cells.

The literature describes the possibility of DNA damage-induced senescence, which prevents the cells from passing on the defective genetic material [74,75]. The CRISPR/Cas9-induced KOs generated could, thus, have triggered senescence due to genome instability. In addition, permanent oxidative stress in the cultures could have led to the induction of senescence. Chronic activation, for example, of the unfolded protein response, leads to oxidative stress up to senescence and also apoptosis of the cells [76,77,78]. Mochitate et al. described reduced ColI and also α-SMA production as a result of stress, which had triggered senescence in the cells [79]. These observations are consistent with the *XYLT2* KO expression pattern. A possible marker of oxidative stress is *NOX4*, whose mRNA expression is significantly induced for *XYLT1* and *XYLT2* KO. *NOX4* can be induced by the shear stress of the cell and, thus, promote the production of ROS and the expression of senescence-associated secretory phenotype [80]. The production of ROS leads to a feed-forward loop in the dysregulation of the ECM. ROS exerts an inhibitory effect on the expression of ColI [81,82]. This mechanism could reinforce the detected changes in the KO models. The initiated oxidative stress in the KOs further promotes the degradation of the ECM and, thereby, reinforces the dysregulation.

Another marker for oxidative stress could be the intracellular Ca^2+^ concentration. The Ca^2+^ stimulates ECM production and, thus, promotes aminopeptidase activity [83,84]. A permanent dysregulation of oxidative stress can lead to apoptosis. An induced caspase3/7 activity for the KO of the *XYLT2* gene was detected in this study. The *XYLT1* KO showed no significant change compared to the control.

Finally, the expressions of exemplary cytokines were examined. The IL1β represses the expression of *ACAN* [85] and is present in the *XYLT1* and *XYLT2* KOs induced at the mRNA level. This could be another explanation for the strongly regulated *ACAN* expression. Both IL6 and IL8 are cytokines of the senescence-associated secretory phenotype [86]. However, IL6 also has a profibrotic effect. Increased IL6 concentrations have been detected in the serum of patients suffering from systemic scleroderma with excessive ECM production [87], and cell culture models of scar formation also showed increased IL6 expression [88]. IL6 can induce ECM expression via the STAT3 signaling pathway [89]. This could be a compensatory mechanism of the cell to compensate for reduced ECM production. The IL6 concentration in the supernatants, as well as IL8 mRNA expression, is elevated for both *XYLT* KOs. This may correlate with the induced senescence and be a mechanism for inducing ECM production.

### 4.6. Characterization of XYLT1- and XYLT2-Double-Deficient Dermal Fibroblasts

For the generation of stable XT-deficient dermal fibroblasts, an attempt was made to perform a sequential KO. Only 5% of the control cell number could be detected in the *XYLT1*/*XYLT2* dKO system 48 h after the completed transfection. Due to the high loss of cells, a stable culture cannot be established. The results indicate that the fibroblasts cannot compensate for a complete XT deficiency. Fischer et al. have already postulated that HEK cells with a complete XT deficiency are not viable [90].

In the next step, an attempt was made to develop an siRNA-mediated knockdown model. Successful knockdown was confirmed both at the mRNA level and due to the reduced XT activity of both isoforms. The mRNA expression of the other glycosyltransferases genes involved in the synthesis of the tetrasaccharide linker were also partially reduced as a result of the dKD. *B4GALT7* and *B3GAT3* were significantly reduced in expression, but *B3GALT6* was unchanged compared to the control. The reduction in *B4GALT7* mRNA expression could not be verified using the GalT-I activity assay, as this was already the case in the individual KOs. A dKD has a similar effect on the enzymes involved in tetrasaccharide linker synthesis as the KOs of the two individual isoforms.

The subsequent characterization of exemplary ECM-associated targets showed an induction of almost all of the genes investigated. The induction of ECM-associated targets is characteristic of myofibroblast differentiation [34]. The induction of ColI (gene: *COL1A1*) detected at the mRNA level was confirmed by immunofluorescence at the protein level. It was noticeable that the collagen fibers formed aggregates. It is possible that these are caused by the probably nonfunctional GAGs and PGs. The latter are required for the assembly of collagen fibrils [91]. Correlating with the increased ColI synthesis, a decreased MMP1 mRNA expression as well as a reduced Pro-MMP1 concentration in the supernatant of the dKD cells compared to the controls could be detected. These observations also indicate myofibroblast differentiation. ColI expression is stimulated and MMP1 expression decreases, for example, through the release of TGFβ1 [92,93].

Increased *ACTA2* and *TGFB1* expression accompanied by a general increase in ECM production is again characteristic of myofibroblast differentiation [42,69,70]. In addition, the XT-deficient fibroblasts showed morphological changes, with the cell bodies appearing larger and forming more lamellipodia. These results are initially surprising, as myofibroblast differentiation was previously associated with induced XT-I activity [94]. One possible explanation could be a compensatory response of the cells to the dysfunctional ECM. The fibroblast induces the increased production of ECM components via various signaling pathways in order to restore the ECM function. Despite reduced XT activity, ECM protein production is induced. A similar reaction has been described in fibroblasts with biglycan deficiency [95].

A possible mechanism for the initiation of ECM production could be the release of TGFβ1 from the ECM of fibroblasts. A dysfunctional ECM structure can release the TGFβ1 stored in latency complexes and, thus induce a feed-forward loop [96]. Therefore, the importance of the TGFβ1 signaling pathway and the formation of focal adhesions for the myofibroblast differentiation observed in this work was analyzed in the next subproject. Fibroblasts were treated with an inhibitor in parallel to dKD to test whether there is a causal relationship between the increased TGFβ1 expression and myofibroblast differentiation during *XYLT1*/*XYLT2* dKD. Care was taken in the inhibitor selection process to ensure that no other kinases were affected. It could be shown that SB431542 treatment inhibits α-SMA expression and, thus there is a connection between myofibroblast differentiation during *XYLT1*/*XYLT2* dKD and the TGFβ signaling pathway. The FAK is another protein involved in myofibroblast differentiation. It transmits signals, for example, mechanical tension emanating from the ECM, via the focal adhesions to various intracellular signaling pathways. The TGFβ signal transduction leads to the activation of the FAK [97]. Reduced α-SMA expression in the *XYLT1*/*XYLT2* dKD fibroblasts was detected as a result of the treatment and a correlation was demonstrated.

The results of the dKD system thus show a short-term myofibroblast differentiation with an accompanying induction of ECM components. In the long term, however, the *XYLT1* and *XYLT2* single KO systems showed reduced ECM component production. One possible starting point for the explanation is the analysis at different time points. Due to the transience, the characterization of the cells after siRNA-mediated knockdown is performed after only a few cell cycles, within 48–72 h. The characterization of the single KOs takes place several cell division cycles after the introduction of the respective KO. The permanent malproduction of ECM molecules and the probable accumulation of dysfunctional proteins trigger stress reactions up to cell cycle arrest in the cells. 

There are a few limitations to this study. So far, we have used a single clone with the corresponding perfect control for each of the two single knockouts. Nevertheless, further runs should be carried out in the future to verify the results of the single clones. This work has demonstrated a method for generating in vitro models of GAG linkeropathies for primary fibroblasts. In the future, this method can be transferred to more long-term cell models, such as immortalized cells. In addition, the analyses of the TGFβ signaling pathway can be transferred to the individual clones in follow-up studies.

## 5. Conclusions

This work demonstrates a method to successfully generate a *XYLT1*- and *XYLT2*-deficient GAG linkeropathy model systems in human dermal fibroblasts. Furthermore, it was possible to perform a complete *XYLT* knockdown. After XT loss, the fibroblasts initially compensate for the misregulated ECM homeostasis with ECM overproduction. Effects were comparable to those observed after TGFβ1-indueced myofibroblast differentiation. The permanent loss of one of the two XT isoforms induces a stress response that leads to the reduction in ECM components. The *XYLT1* KO still allows α-SMA overproduction. Cells may be able to partially compensate for the loss via unaltered XT-II activity. The *XYLT2* KO leads to a reduction in both XT isoforms and a strong stress response in case of permanent dysregulation. In addition to oxidative stress, induced senescence and evidence of apoptotic cells has been proven. 

There are temporal- and isoform-specific regulatory differences in the *XYLT* deficiency models investigated here, which need to be analyzed in more detail in further studies. 

## Figures and Tables

**Figure 1 biomedicines-12-00572-f001:**
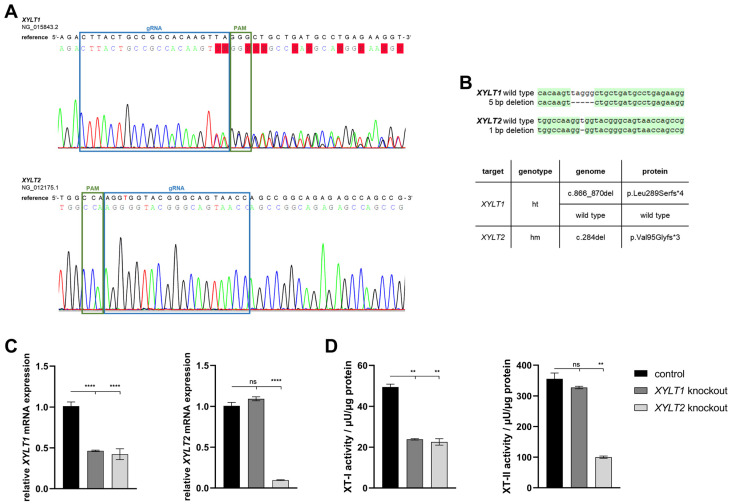
Characterization of *XYLT1* or *XYLT2* knockout (KO) fibroblasts using an RNP-based CRISPR/Cas9 system. (**A**) Comparison of the genomic DNA sequences of the KO clones to their respective reference sequence. The blue box marks the binding region of the gRNA (*XYLT1*: g 211908-211930; *XYLT2*: g 7647-7669). The PAM is indicated in green. (**B**) Sequence alignment after TA cloning to identify the mutations and the corresponding names of the clones, as well as the effects on the amino acid sequence (AS) of the XT-I and XT-II proteins (ht = heterozygous; hm = homozygous). (**C**) Analysis of the *XYLT1* and *XYLT2* mRNA expression of the KO fibroblasts is shown. Control (n = 1) and both KO cultures (n = 1) were seeded at a cell density of 50 cells/mm^2^. Relative gene expression analysis was performed after 72 h of cultivation using the ΔΔC_T_ method for the *XYLT1* and *XYLT2* genes. The expression of *RPL13A*, *SDHA* and *B2M* genes was used for normalization. (**D**) Analysis of intracellular XT-I and -II activity was performed after 72 h of cultivation using the SPE-UPLC-MS/MS method. Mean values with standard errors shown are composed of three biological and three technical replicates each (ns = not significant; ** *p* < 0.01; **** *p* < 0.0001).

**Figure 2 biomedicines-12-00572-f002:**
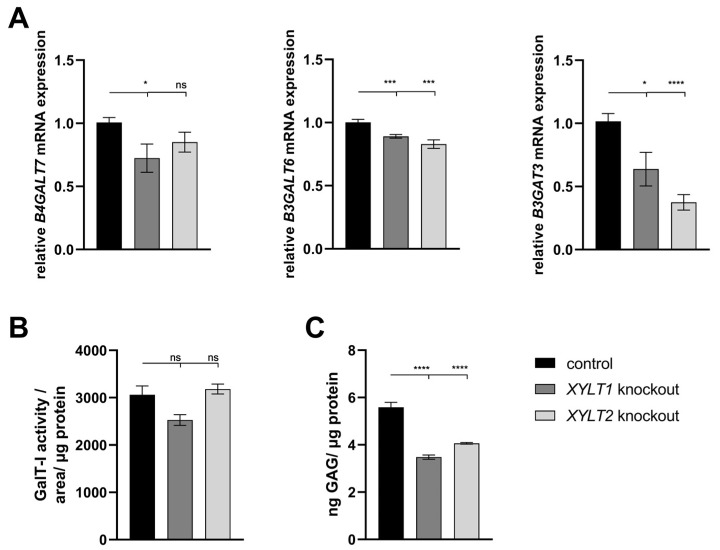
Analysis of glycosyltransferase, as well as sulfated glycosaminoglycan (sGAG) expression of KO fibroblasts. (**A**) The control (n = 1) and the two KO cultures (n = 1) were seeded at a cell density of 50 cells/mm^2^ and analyzed by the ΔΔC_T_ method after 72 h of cultivation for the analysis of *B4GALT7*, *B3GALT6* and *B3GAT3* mRNA expression. The expression of *RPL13A*, *SDHA* and *B2M* genes was used for normalization. (**B**) Analysis of the GalT-I activity was also performed after 72 h of cultivation using the SPE-UPLC-MS/MS method. Normalization was performed to the total protein concentration. (**C**) Cells were seeded at a cell density of 80 cells/mm^2^ and lysed and isolated by papain after 144 h for the determination of sGAG concentration. Normalization was performed to the total protein concentration. Mean values with standard errors shown are composed of three biological and three technical replicates each (ns = not significant; * *p* < 0.1; *** *p* < 0.001; **** *p* < 0.0001).

**Figure 3 biomedicines-12-00572-f003:**
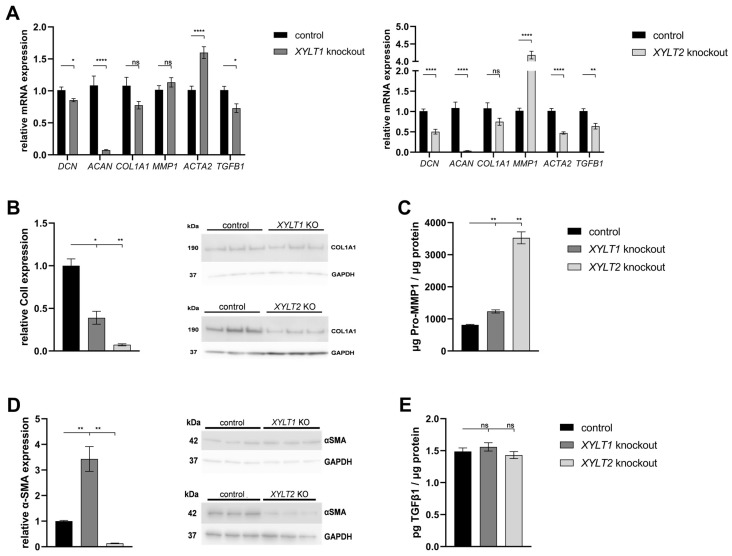
Analysis of extracellular matrix (ECM)-associated genes and proteins from *XYLT1* or *XYLT2* KO fibroblasts. (**A**) The control (n = 1) and the two KO cultures (n = 1) were seeded at a cell density of 50 cells/mm^2^ and analyzed by the ΔΔC_T_ method after 72 h of cultivation for the analysis of *DCN*, *ACAN*, *COL1A1*, *MMP1*, *ACTA2* and *TGFB1* mRNA expressions. The expression of *RPL13A*, *SDHA* and *B2M* genes was used for normalization. Means with standard errors presented are composed of three biological, as well as three technical, replicates each. Quantification of ColI (**B**) and α-SMA (**D**) was performed by Western blot after 144 h of cultivation. Quantification was performed by ImageJ and GAPDH expression was used for normalization. Means with standard errors presented are composed of three biological replicates. Pro-MMP1 (**C**) and TGFβ1 (**E**) expressions were quantified by ELISA and normalized to total protein concentration. Means with standard errors presented are composed of three biological and as three technical replicates each (ns = not significant; * *p* < 0.1; ** *p* < 0.01; **** *p* < 0.0001).

**Figure 4 biomedicines-12-00572-f004:**
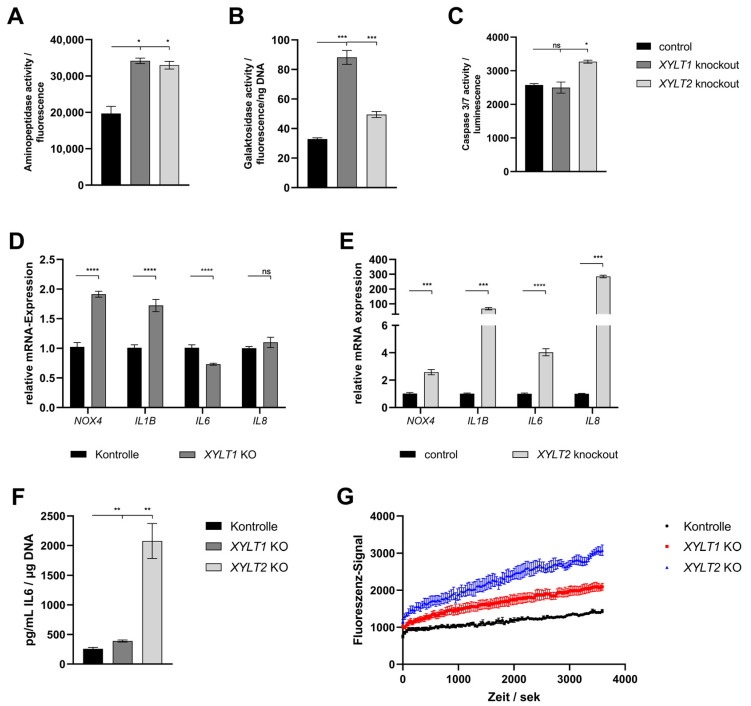
Analysis of different markers for the analysis of viability, senescence, apoptosis and oxidative stress in *XYLT1* or *XYLT2* KO fibroblasts. (**A**) Cells were seeded at a cell density of 70 cells/mm^2^ and the aminopeptidase activity of four biological replicates was determined after 72 h for the quantification of viability. (**B**) The cellular senescence was determined by SA β-galactosidase activity after 72 h from three biological and two technical replicates. Normalization was performed by reference to the total DNA concentration. (**C**) Apoptosis quantification after 74 h was performed using caspase 3/7 activity. This was based on three biological replicates. (**D**,**E**) The control and *XYLT1* and *XYLT2* KO fibroblasts were seeded at a cell number of 50 cells/mm^2^ and analyzed after 72 h using the ΔΔC_T_ method for the relative gene expression analysis of *NOX4*, *IL1B*, *IL6* and *IL8*. *RPL13A*, *SDHA* and *B2M* were used as housekeeping genes. The mean values with standard errors presented are composed of three biological and three technical replicates. (**F**) The IL6 protein expression was quantified by ELISA after 72 h in the supernatant of the cultures. The mean values and the standard error are composed of three biological replicates and were related to the total DNA of the samples. (**G**) An amount of 250 cells/mm^2^ were seeded and the uptake of Fluo-8 was analyzed in six replicates each after 24 h to determine the intracellular Ca^2+^ concentration (ns = not significant; * *p* < 0.1; ** *p* < 0.01; *** *p* < 0.001; **** *p* < 0.0001).

**Figure 5 biomedicines-12-00572-f005:**
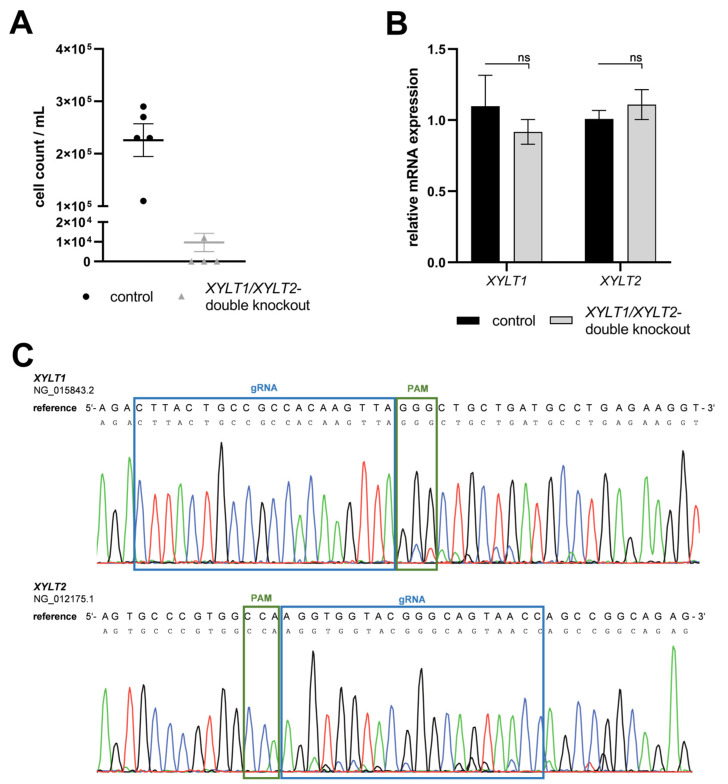
Characterization of an *XYLT1* and *XYLT2* double KO (dKO) in dermal fibroblasts. (**A**) Normal human dermal fibroblasts (NHDFs) (n = 1) were seeded at a cell density of 50 cells/mm^2^ in gelatin-coated six-well plates and transfected Lipofectamine 2000 mediated with 3.8 nM *XYLT1* and 3.8 nM *XYLT2*-Cas9-RNP complex for the dKO,. The cell number of viable cells was determined by tryptan blue 48 h after the second transfection. (**B**) Relative gene expression analysis was performed 72 h after the second transfection for the *XYLT1* and *XYLT2* genes. (**C**) The dKO genomic DNA was isolated 24 h after the second transfection and analyzed by Sanger sequencing. The region of gRNA (blue box; g 211908-211930; XYLT2: g 7647-7669) and PAM (green box) is highlighted. Means and standard errors shown are composed of three biological and three technical replicates (ns = not significant).

**Figure 6 biomedicines-12-00572-f006:**
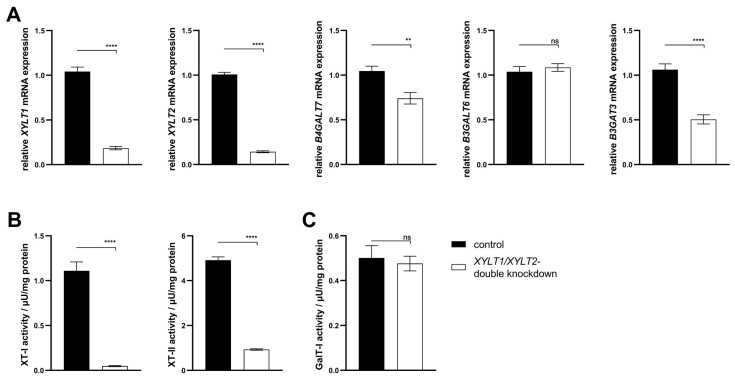
Analysis of the gene expression of *XYLT1*, *XYLT2*, *B4GALT7*, *B3GALT6* and *B3GAT3* and the activity determination of XT-I, XT-II and GalT-I. (**A**) The NHDFs (n = 4) were seeded at a cell density of 119 cells/mm^2^ and transfected Lipofectamine 2000 mediated with 50 nM *XYLT1* and 50 nM *XYLT2* siRNA. As a control, NHDFs were treated with 100 nM of a control siRNA. A relative gene expression analysis was performed after 48 h of cultivation using the ΔΔC_T_ method for the *XYLT1*, *XYLT2*, *B4GALT7*, *B3GALT6* and *B3GAT3* genes. The expression of *RPL13A* and *GAPDH* genes were used for normalization. Analysis of XT-I and -II activity (**B**), as well as GalT-I activity (**C**) was performed after 72 h of cultivation using the SPE-UPLC-MS/MS method. Mean values with standard errors shown are composed of three biological, as well as three technical, replicates each (ns = not significant; ** *p* < 0.01; **** *p* < 0.0001).

**Figure 7 biomedicines-12-00572-f007:**
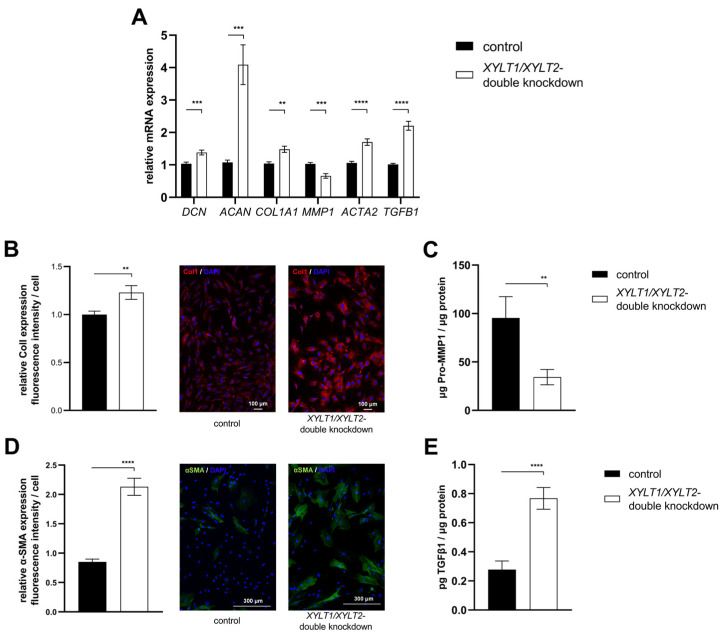
Analysis of ECM-associated genes and proteins of *XYLT1*/*XYLT2* double-knockdown (dKD) fibroblasts. (**A**) The NHDFs (n = 4) were seeded at a cell density of 119 cells/mm^2^ and transfected Liofectamine 2000 mediated with 50 nM *XYLT1* and 50 nM *XYLT2* siRNA. The NHDFs were treated with 100 nM of a control siRNA as a control. Relative gene expression analysis was performed after 48 h of cultivation using the ΔΔC_T_ method. The genes analyzed were *DCN*, *ACAN*, *COL1A1*, *MMP1*, *ACTA2* and *TGFB1*. *RPL13A* and *GAPDH* were used as housekeeping genes. The mean values with standard errors presented are composed of three biological, as well as three technical, replicates each. (**B**) Quantification of ColI was performed by immunofluorescence. Cells were fixed with 4% paraformaldehyde 72 h after the dKD. Staining was performed by a primary rabbit anti-ColI, secondary goat anti-rabbit (Alexa-550-labeled) antibody (red) and DAPI (blue). The quantification of total fluorescence from six images each of three biological replicates is shown. Fluorescence was normalized to the cell number of each image. (**C**) The supernatant was analyzed after 72 h of cultivation and normalized to the respective total protein amount for the Pro-MMP1 ELISA. The mean with the standard error from three biological and two technical replicates is shown. (**D**) Quantification of α-SMA was performed by immunofluorescence. Cells were fixed with acetone/methanol (1:1, *v*/*v*) 72 h after the dKD. Staining was performed by primary mouse anti-α-SMA, secondary goat anti-mouse (Alexa-488-labeled) antibody (green) and DAPI (blue). The quantification of total fluorescence from six images each of three biological replicates is shown. Fluorescence was normalized to the cell number of each image. (**E**) The supernatant was analyzed after 72 h of cultivation and normalized to the respective total protein amount for the TGFβ1 ELISA. The mean with standard error from three biological and two technical replicates is shown (** *p* < 0.01; *** *p* < 0.001; **** *p* < 0.0001).

**Figure 8 biomedicines-12-00572-f008:**
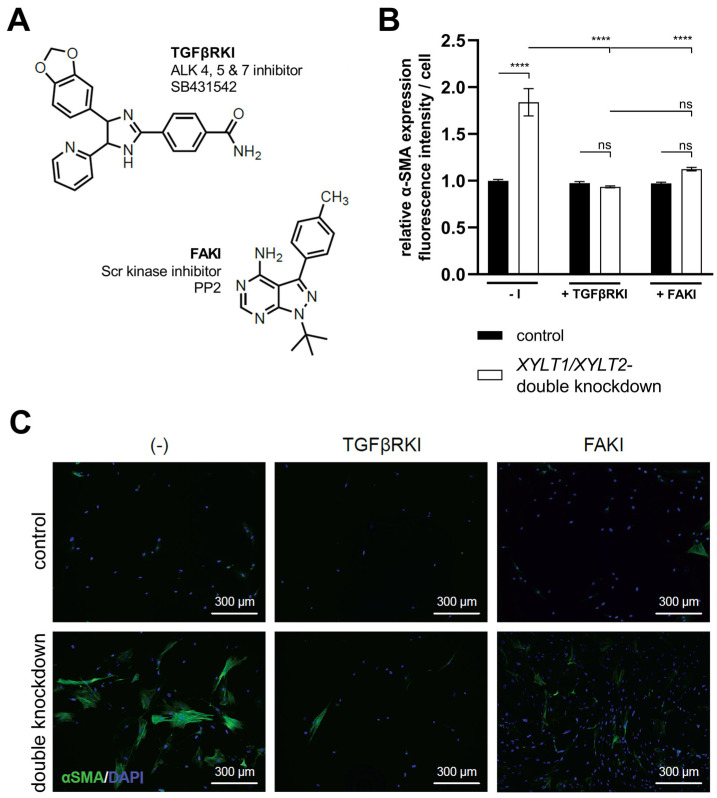
Quantification of α-SMA in *XYLT1/XYLT2* dKD fibroblasts after treatment with TGFβRKI or FAKI. The NHDFs (n = 1) were seeded at a cell density of 50 cells/mm^2^ on collagen-coated chamber slides and transfected Lipofectamine 2000 mediated with 50 nM *XYLT1*- and 50 nM *XYLT2*-siRNA. As a control, NHDFs were treated with 100 nM of a control siRNA. Either 10 μM TGFβRKI, 10 μM FAKI or dimethyl sulfoxide (−I) was added to the lipofection medium. After 72 h, acetone/methanol (1:1, *v*/*v*) fixation of the cells was performed for the analysis of α-SMA protein expression by immunofluorescence analysis. Staining was carried out using primary mouse anti-α-SMA, secondary goat anti-mouse (Alexa-488-labeled) antibody (green) and DAPI (blue). The Lewis structures of the inhibitors used (**A**), quantification of total fluorescence from six images of each of three biological replicates (**B**) and exemplary images at 100× magnification are shown (**C**) (ns = not significant; **** *p* < 0.0001).

**Table 1 biomedicines-12-00572-t001:** crRNA for the CRISPR/Cas9 KO of the XT genes. The nucleotides were predesigned and synthesized by IDT (USA).

Target	5′–3′ Sequence	IDT Label
*XYLT1*	CTTACTGCCGCCACAAGTTA	*XYLT1.1.AA*
*XYLT2*	GGTTACTGCCCGTACCACCT	*XYLT2.1.AB*

**Table 2 biomedicines-12-00572-t002:** siRNA sequences for the siRNA-mediated knockdown. The siRNAs were predesigned and synthesized by ThermoFisher (Waltham, MA, USA).

Target	5′–3′ Sequence
*XYLT1*	GCAUCAUGCUACCAAUCUGtt ttCGUAGUACGAUGGUUAGAC
*XYLT2*	GGCCGUUUAUCACGAGCAGtt ttCCGGCAAUAGUGCUCGUC

**Table 3 biomedicines-12-00572-t003:** Primer sequences, annealing temperatures (TA) and resulting product sizes used for qPCR.

Target	Sequence	TA/°C	Product Size/bp
*ACAN*	CACCCCATGCAATTTGAG GCCACTGTGCCCTTTTTA	63	158
*ACTA2*	GACCGAATGCAGAAGGAG CGGTGGACAATGGAAGG	59	169
*B2M*	TGTGCTCGCGCTACTCTCTCTT CGGATGGATGAAACCCAGACA	63	137
*B3GALT6*	CCCCGCTGTGGTCTTTGTTG CGCCCCCGTTTCTTCCTC	63	188
*B3GAT3*	GCTGTTTGAGGAGATGCGCTG TCAGAAGACTGCTCTCCAGGT	63	251
*B4GALT7*	GCCATGCACAGTGATCAGAG CCCTACACTGTGTCTCTGCA	63	196
*COL1A1*	GATGTGCCACTCTGACT GGGTTCTTGCTGATG	63	151
*DCN*	CCTTCCGCTGTCAATG GCAGGTCTAGCAGAGTTG	63	102
*GAPDH*	AGGTCGGAGTCAACGGAT TCCTGGAAGATGGTGATG	59	223
*IL1B*	ACAGATGAAGTGCTCCTTCCA GTCGGAGATTCGTAGCTGGAT	63	73
*IL6*	ACAGCCACTCACCTCTTCAG GTGCCTCTTTGCTGCTTTCAC	63	122
*IL8*	GAACTGAGAGTGATTGAGAGTGGA CTCTTCAAAAACTTCTCCACAACC	63	125
*MMP1*	AGAAACACAAGAGCAAGATGTG TGGCGTGTAATTTTCAATCCTGT	63	298
*NOX4*	CTTCCGTTGGTTTGCAGATT GAATTGGGTCCACAACAGA	63	246
*RPL13A*	CGGAAGGTGGTGGTCGTA CTCGGGAAGGGTTGGTGT	63	115
*SDHA*	AACTCGCTCTTGGACCTG GAGTCGCAGTTCCGATGT	63	177
*TGFB1*	GCGATACCTCAGCAACC ACGCAGCAGTTCTTCTCC	63	331
*XYLT1*	GAAGCCGTGGTGAATCAG CGGTCAGCAAGGAAGTAG	63	281
*XYLT2*	ACACAGATGACCCGCTTGTGG TTGGTGACCCGCAGGTTGTTG	63	139

**Table 4 biomedicines-12-00572-t004:** The antibodies and concentrations used for Western blotting.

Primary Antibody	Dilution	Secondary Antibody	Dilution
*Rabbit-anti-COL1A1* (ab34710; Abcam, Cambridge, UK)	1:10,000	*Goat-anti-rabbit IgG-HRP polyclonal* (ab150113; Abcam, Cambridge, UK)	1:5000
*Mouse-anti-αSMA* (GA61161-2; Cell Signaling Technology, Cambridge, UK)	1:1000	*Horse-anti-mouse igG-HRP polyclonal* (ab150078; Abcam, Cambridge, UK)	1:2000
*Mouse-anti-gapdh* (ab8245; Abcam, Cambridge, UK)	1:5000	*Horse-anti-mouse igG-HRP polyclonal* (ab150078; Abcam, Cambridge, UK)	1:2000

**Table 5 biomedicines-12-00572-t005:** The antibodies and concentrations used for immunohistochemical staining.

Primary Antibody	Dilution	Secondary Antibody	Dilution
*Rabbit-anti-COL1A1* (ab34710; Abcam, Cambridge, UK)	1:100	*Goat-anti-rabbit (Alexa-555)* (A-32732; Thermo Fisher Scientific, USA)	1:200
*Mouse-anti-αSMA* (GA61161-2; Cell Signaling Technology, Cambridge, UK)	1:50	*Goat-anti-mouse (Alexa-488)* (A-11001; Thermo Fisher Scientific, USA)	1:200

## Data Availability

The original raw data and materials presented in the study can be made available upon request. Further inquiries can be directed to the corresponding author.

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
