# Peer review of "Xylosyltransferase-Deficiency in Human Dermal Fibroblasts Induces Compensatory Myofibroblast Differentiation and Long-Term ECM Reduction"

_biomedicines, 2024, doi:10.3390/biomedicines12030572_

Round 1

Reviewer 1 Report

Comments and Suggestions for Authors

Authors tried to understand the pathophysiology of the Desbuquois dysplasia type 2, which can be caused by heterozygous defects in xylosyltransferase 1 (XYLT1) gene. For this aim, they developed XYLT1 haploinsufficient human fibroblasts by using a CRIPR/Cas9 system and showed that XYLT1 deficiency caused inappropriate induction of alpha-SMA gene, suggesting a differentiation shift into myofibroblasts by XYLT1 insufficiency. 

They also showed the effects of CRIPR/Cas9-mediated XYLT2 deficiency (probably due to homozygous XYLT2 defects) and XYLT1/XYLT2 knockdown by siRNA introduction in human fibroblasts. However, XYLT2 is not a causal gene for Desbuquois dysplasia type 2, and therefore, the presentation of those data in main text induces confusion in the mind of readers. 

In addition, the data regarding ACAN is confusing: ACAN gene expression was strongly suppressed in both XYLT1- and XYLT2-deficient cells (Figure 3A), while its expression was remarkably upregualted in XYLT1/XYLT2 knockdown cells (Figure 7A).

Due to these confusing points, the reviewer does not recommend the presentation of the data regarding XYLT2-deficient cells and those of XYLT1/XYLT2 knockdown cells in main text.  

The reason why authors took those "unrelated" approaches may come from the fact that they could not detect the induction of TGFB1 gene in XYLT1 haploinsufficient human fibroblasts. Nevertheless, they successfully showed TGFB1 gene induction in XYLT1/XYLT2 double knockdown cells. However, XYLT1/XYLT2 double knockdown does nor properly reflect the genetic background of Desbuquois dysplasia type 2 patients. 

Their hypothesis that "XYLT1 deficiency inappropriately induces myofibroblast differentiation via TGFB1 signaling" is very interesting and has novelty. 

It is well known that an activation of TGFB1 signaling is induced not only by its gene induction but also the release of the protein from its latent form, which consist of the LAP/LTBP/TGFB complex. To check the activation of TGFB1, immunostaining study using an anti-phosho-smad2/smad3 antibody is useful. If nuclear translation of phosho-smad2/smad3 is detected, it indicates that TGFB signaling is activated in the cells.

If authors can show the activation of TGFB1 in XYLT1 haploinsufficient human fibroblasts, the point of view will be simplified. In that case, the data regarding XYLT2-deficient and XYLT1/XYLT2 knockdown cells can be translocated to supplemental information to avoid confusion of the readers mind.

Major concerns:

1. To assess the possible activation of TGB signaling by XYLT1 deficiency, please perform immunostaining study using an anti-phosho-smad2/smad3 antibody in XYLT1 haploinsufficient human fibroblasts.

2. In Title, the word "acute" sounds rather bizzar since this term indicates the rapidness in the process of disease development in patients. Please change the title into a proper one.

Minor concerns:

In line 64, the word "ambylophilia" should be corrected as amblyopia.

Reviewer 2 Report

Comments and Suggestions for Authors

Major issues:

Several pitfalls in study design are apparent:

1)     The work is based on single clones so is unknown if defects are really due to the mutations introduced (Are rescue experiments possible? Is comparison with transient silencing possible as double KO was achieved only by silencing?)

2)     For XYLT1 only an heterozygous mutant was obtained, while in the XYLT2 KO a reduction in XYLT1 was observed making results interpretation challenging.  

3)     In the introduction discuss available disease models such as XYLT1/2 ko mice and patients at the level of molecular defects. This would help to understand the logic behind the selection of putative targets examined, a lot of interesting changes are presented but their mechanistic link to XYLT1/2 is not clear to the reader (also some graphics/scheme could help in this). A good logic is presented in the discussion AFTER the data.

4)     Are the fibroblasts used immortalized? Otherwise, the claim of having generated a model of the disease is overstated, a method (low efficiency) for the generation of such models is the actual result.

Minor issues

In Fig. 7 label the images to allow the reader understand who is who.

The rescue experiments with TGFbeta inhibitors are very interesting but done only for the double silenced cells, could them be extended to XLYT2 KO?

Recommendations:

1)     Improve introduction and insert a graphical abstract/summary.

2)     Compare the ko lines with silenced cells.

3)     Revise the claims about model generation.

Reviewer 3 Report

Comments and Suggestions for Authors

The authors provide an interesting paper about knocking out the XYLT genes. The studies are well done, and the results are convincing. My only complaint is that the discussion is extremely long and tends to repeat the findings reported in the results. I suggest that the authors shorten the paper, especially the discussion, to be more to-the-point. 

Round 2

Reviewer 1 Report

Comments and Suggestions for Authors

In the revised manuscript as well as point-by-point responses, authors have sufficiently addressed the concerns raised by the reviewer. There, the current manuscript has been sufficiently improved to warrant publication in biomedicines.

Reviewer 2 Report

Comments and Suggestions for Authors

Minor text changes were done, my perplexities on the experimental approach remain unchanged and preclude publication.
